# Mass bathing events in River Kshipra, Central India- influence on the water quality and the antibiotic susceptibility pattern of commensal *E.coli*

**Manju Purohit**[1,2‡]*, **Vishal Diwan**[1,3,4‡], **Vivek Parashar**[3], **Ashok J. Tamhankar**[1,5], **Cecilia Stålsby Lundborg**[1]

**1** Department of Public Health Sciences, Karolinska Institutet, Stockholm, Sweden, **2** Department of Pathology, R.D. Gardi Medical College, Ujjain, India, **3** ICMR-National Institute for Research in Environmental Health, Bhopal, Madhya Pradesh, India, **4** Department of Public Health and Environment, R. D. Gardi Medical College, Ujjain, India, **5** Department of Environmental Medicine, Indian Initiative for Management of Antibiotic Resistance, R.D. Gardi Medical College, Ujjain, India

‡ These authors share first authorship on this work.
* manjuraj.purohit@ki.se

## Abstract

### Background

Antibiotic resistance is one of the major global health emergencies. One potential source of dissemination of resistant bacteria is mass gatherings, e.g. mass bathing events. We evaluated the physicochemical parameters of water quality and the antibiotic resistance pattern in commensal *Escherichia coli* from river-water and river-sediment in pre-, during- and post-mass bathing events in river Kshipra, Central India.

### Method/Design

Water and sediment samples were collected from three selected points during eight mass bathing events during 2014–2016. Water quality parameters (physical, chemical and micro-biological) were analyzed using standard methods. In river water and sediment samples, antibiotic susceptibility patterns of isolated *E. coli* to 17 antibiotics were tested.

### Results

pH, turbidity and dissolved oxygen were significantly lower and total dissolved solid, free carbon dioxide were higher during mass bathing, whilst TSS, BOD and COD were lowest in pre-bathing and highest in post-bathing period. *E.coli* with multi drug resistance (MDR) or extended spectrum beta-lactamase (ESBL) production were between 9–44% and 6–24%, respectively in river-water as well as river-sediment. Total coliform count/ml and *E. coli* count were higher during-and post-bathing in river water than in pre-bathing period. Thus, the percentage of resistance was significantly higher during and post-bathing period (*p*<.05) than in pre-bathing. Colony forming unit (CFU)/ml in river-sediment was much higher than in

**Data Availability Statement:** All relevant data are within the manuscript and its Supporting Information files.

**Funding:** The project was funded by Swedish Research Council (grant no 521-2012-2889) and 2017- 01237. The funders had no role in study design, data collection and analysis, decision to publish, or preparation of the manuscript.

**Competing interests:** The authors have declared that no competing interests exist.

river-water. Percentage of resistance was significantly higher in river-water ($p$<.05) than in river-sediment.

## Conclusions

Antibiotic resistance in *E.coli* isolated from the Kshipra River showed significant variation during mass bathing events. Guidelines and regulatory standards are needed to control environmental dissemination of resistant bacteria.

## Introduction

Clean, pure and safe water exists briefly in nature, as it is immediately polluted by prevailing environmental factors and anthropogenic activities. Industrialization and urbanization has polluted rivers mostly via agricultural runoffs and industrial effluents containing many used or unused antibiotics [1]. Antibiotics alter the ecology of the environment and generate antibiotic resistance [2]. It has been shown that the accumulation of antibiotics in the environment facilitate bacterial adaptation response to develop antibiotic resistant genes [3, 4].

The organic pollution of river-water and at river banks is an increasing problem of worldwide concern [5–7]. *Escherichia coli (E. coli)*, usually a commensal coliform of humans and animals, enters into the river mainly from livestock operations, their waste products and human septate [8]. These bacteria may accumulate in the river-sediment and might disseminate to distant sites with the antibiotic resistant genes acquired through horizontal transfer of genetic material. The presence of antibiotic resistant bacteria in freshwater sources has been documented throughout the world [4, 9, 10].

In India, river-water is considered sacred and mass bathing in some rivers is an age-old ritual. River Kshipra, one of the sacred rivers of India, is a source of domestic water supply for Ujjain City and around. Due to the religious importance of Kshipra River, mass gatherings including mass bathing of large or small scales occur throughout the year at its bank. Mass gathering is a pre-planned assemblage of more than 1000 persons at a particular place for a certain period [11]. During these mass gathering events, pilgrims from all over the country visit Ujjain for bathing in river. People take a 'holy-dip' as a part of religious ritual in this river in the form of mass bathing. They not only take 'holy-dip' in the river-water for bathe but also drink a handful of 'holy-water'. There is a high possibility of ingestion of antibiotic resistant bacteria when they bathe in the river and/or drink river water. Thus, the pilgrims are exposed to antibiotic resistant genes and disseminate it to the wider world.

Occurrence of antibiotic resistant bacteria has been reported from different rivers of India [12–14]. It is important to develop knowledge about the role and dynamicity of presence of antibiotic resistance in river-water during mass gatherings to design approaches to contain antibiotic resistance in rivers. In general, there is a paucity of information on the presence and concentration of antibiotic resistance in rivers associated with mass bathing in India. The influence of mass gathering and associated mass bathing on the antibiotic resistance pattern in bacteria in the water and sediment of river Kshipra, Central India, is not studied. Thus, our aim wasto compare and correlate the antibiotic susceptibility pattern and total burden of commensal *E.coli* from river-water and river-sediment in pre-, during- and post- mass bathing events in river Kshipra. We further aimed to evaluate the physicochemical parameters of water quality during these time events.

## Materials and methods

### Study setting and sample collection

The present study is a part of an ongoing project, which has been described in detail previously [15]. In brief, the study samples were collected from River Kshipra, in its 93 km flow through Ujjain district, before, during and after mass-bathing events. Approximately 50,000–200,000 pilgrims take 'holy-dip' during any mass bathing event. Samples of river-water and river-sediment were collected from pre-identified and described sites (Fig 1) at pre-, during-, and post-bathing time during eight mass bathing events that occurred between August -2014 to July 2016 (the sites were selected as they have the highest number of pilgrim visits for mass bathing).

The pre-bathing samples were collected one day prior of the bathing event, post-bathing samples were collected 6–7 hours after bathing event while during-bathing samples were collected at the peak hours of the bathing event, generally between 9 to 11AM. Each time, river-water and -sediment were collected in sterile screw-capped bottles and conical falcon tube respectively as described in detail [15]. Approximately 2Kg of sediment and 3.5L river-water

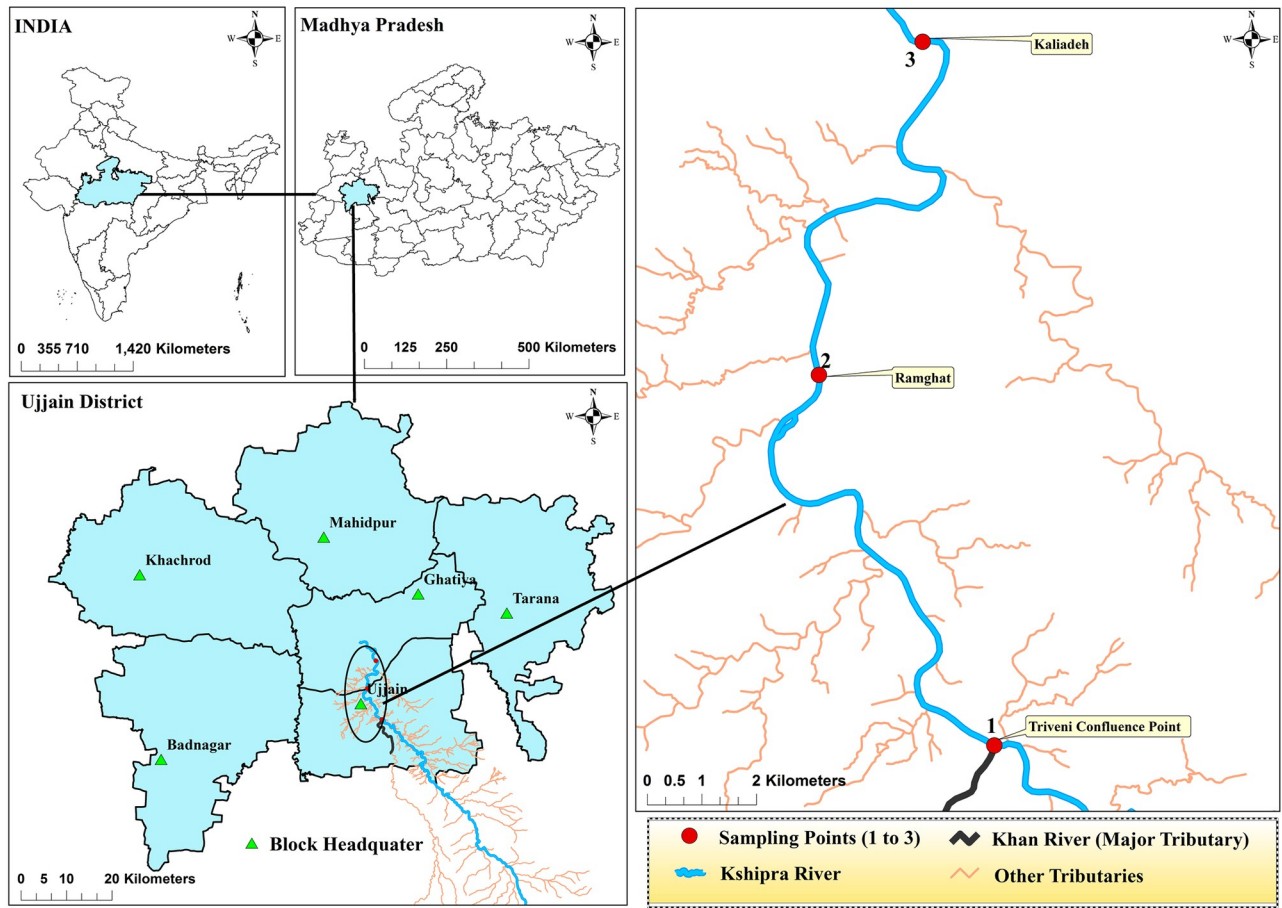

**Fig 1. Geographical location of the river Kshipra sites for mass bathing sampling.** River Kshipra, River Khan, others tributaries and locations of sampling points were obtained from base map of ArcInfo and Google image and through field Survey. Shp-files were generated using ArcMap version 10. Map boundaries of India, Madhya Pradesh and districts were taken from GADM database (https://gadm.org/) where these are available freely for academic purposes. Jammu and Kashmir and Ladakh were treated as one administrative division as the boundary maps separating the two were not available in retrieved databases.

were collected for analysis of water quality parameters, colony counts and antibacterial suscep-tibility as described elsewhere [15]. All the collected samples were stored in icebox at 4–6°C and transported to central research laboratory of R D Gardi Medical College, Ujjain within one hour.

Ethical permission for the study was obtained from the Institutional Ethics Committee of R D Gardi Medical College, Ujjain, MP, India (No 2013/07/17-311). No special permission was required for water and sediment samples as it was taken only from public land.

## Physico-chemical and biological parameters

The physical and chemical parameters (ambient and water temperature, pH, conductivity, total dissolved solids (TDS) free carbon dioxide (free $CO_2$), carbonate alkalinity and dissolved oxygen (DO)) of collected river-water and river-sediment samples were analyzed in field [15]. After transport to the laboratory, samples were analyzed for turbidity, hardness, chloride, alka-linity, nitrate nitrogen available phosphorous, total suspended solids (TSS), biochemical oxy-gen demand (BOD), chemical oxygen demand (COD) and total phosphorous [15].

## Microbiological and molecular methods

The samples were processed as described previously [15] for (a) Bacterial enumeration- Ten-fold serial dilutions (1:100, 1:1000 as per turbidity of sample) of river-water were done with 0.9% normal saline. The river-sediment sample was first added to 100ml 0.9% normal saline and then ten-fold serial dilutions were performed. All diluted samples were processed follow-ing standard membrane filtration technique using 47mm diameter and pore size 0.45μm membrane filters. After filtration, the membranes were directly inoculated on selective and differential media HiCrome Coliform Agar for 24 hours at 37°C for cultivation, isolation and identification of *E. coli* (blue-violet colony) and non *E. coli* isolates. The estimation of total coliform and total *E. coli* count in colony forming unit (CFU)/100ml was performed, (b) Iden-tification of six *E. coli* isolates per river-water and -sediment sample was done followed by isolation and confirmation of these isolates by PCR and (c)The confirmed *E. coli* isolates were tested for antibiotic susceptibility to nine commonly used classes of antibiotics (cephalospo-rins; quinolones; nalidixc acid; tetracyclins; penicillins; carbapenems; aminoglycosides; sulpha drugs; polymyxins) by Kirby Bauer disc diffusion test on Muller Hinton (MH) Agar with the bacterial suspension of 0.5 McFarland turbidity. The zone diameter of bacterial growth inhibi-tion was interpreted as per Clinical and Laboratory Standard Institute guidelines [16]. The results were interpreted as—number of resistant or susceptible isolates per antibiotic type per sample (out of six isolates), production of beta-lactamase (only where beta-lactamase produc-tion is indicated as a possible mechanism explaining observed resistance) by testing the com-bined disc diffusion method(isolates resistant to either ceftazidime or ceftriaxone (third generation cephalosporin), the presence of co-resistance (phenotypic resistance to two antibi-otics of same or different group per isolate), multiple drug resistance (phenotypic resistance to more than two antibiotics of same or different group per isolate), and multi-drug resistance (MDR) (co-resistance involving three or more antibiotics of three different groups) in each isolate. *E. coli* reference strain ATCC 25922 was used for quality control. Intermediate resistant isolates were categorized as resistant.

The total bacterial DNA from *E. coli* isolates was extracted using the alkaline lysis method. The genetic confirmation of *E. coli* was done through PCR with genus-specific oligonucleotide primers [17]. The confirmed *E.coli* were tested for the presence of various genes (*mcr-1*, *chuA*, *yjaA* and *TspE4C2*) for detecting colistin resistance and phylogenetic grouping as mentioned in detail previously [10, 18].

## Data management and analysis

A pilot study was conducted prior to the main study to train research assistants for field and laboratory work and validate the sampling and analysis methods. All results were recorded in a single dataset using bar codes for unique identification and track of samples at the field and within the laboratories. All data were double-checked for quality assurance, cleaned and entered in to IBM SPSS Statistics 23.0 (SPSS Inc., Chicago, IL, USA). Results were noted for the variation in coliform load (in terms of CFU per 100ml) and antibiotic susceptibility in river-water and -sediment were analyzed and compared between different time-points during events and sites. A paired *t test* was applied for comparing the means of the water quality parameters between pre-, during-, and post-bathing sessions. A *z-test* for difference of proportion was applied for comparing the antibiotic susceptibility pattern to selected antibiotics in the pre-, during, and post- and the first, second, and third bathing sessions. A significant association determined by p-values < 0.05.

## Results

### Study samples

Samples were collected from eight important mass bathing events during the study period as shown in Table 1. Total 144 river-water and 72 river-sediment samples were collected and totally 807/864 and 353/486 *E. coli* were finally isolated, tested and analyzed from river-water and river-sediment respectively. Six river-water (two pre- and four during-bathing) samples showed no growth of *E.coli*.

### Physico-chemical properties of water

Various water quality parameters showed significant difference in during- and/or post-bathing samples as compared to pre-bathing samples (Table 2). Ambient temperature, pH, turbidity and dissolved oxygen were significantly lower and total dissolved solid, free carbon dioxide were higher during-bathing whilst TSS, BOD and COD were lowest in pre-bathing and highest in post-bathing period. Similarly, total and organic phosphorus were highest in post-bathing samples.

### CFUs of coliform and *E. coli*

The *E. coli* CFU/ml for the three sites during all events in river-water was between $0.09x10^3$-$51x10^3$ (mean $5.2x10^3$) and in river-sediment $2.2x10^3$-$9590x10^3$ (mean $474x10^3$) whiles the

**Table 1. Number of samples collected during mass-bathing events took place in river Kshipra, Central India during study period.**

| Month/year of event | Number of sample RW/RS* |
| --- | --- |
| August 2014 | 18/9 |
| November 2014 | 18/9 |
| April 2015 | 18/9 |
| May 2015 | 18/9 |
| October 2015 | 18/9 |
| November 2015 | 18/9 |
| January 2016 | 18/9 |
| February 2016 | 18/9 |

*RW-river-water; RS-river-sediment

**Table 2. Mean value of various physico-chemical parameters of water samples collected at each time points in all mass-bathing events in river Kshipra, Central India.**

|  | Event | Mean | p-value |
|---|---|---|---|
| Ab temp (0C) | Pre | 31.9 | 0.00 |
|  | During | 29.1 |  |
|  | Pre | 31.9 | 0.07 |
|  | Post | 32.3 |  |
|  | During | 29.1 | 0.00 |
|  | Post | 32.3 |  |
| pH | Pre | 8.58 | 0.00 |
|  | During | 8.29 |  |
|  | Pre | 8.6 | 0.00 |
|  | Post | 8.34 |  |
|  | During | 8.30 | 0.4 |
|  | Post | 8.34 |  |
| TDS | Pre | 742.6 | .25 |
|  | During | 755.2 |  |
|  | Pre | 742.6 | .53 |
|  | Post | 750.5 |  |
|  | During | 755.2 | .47 |
|  | Post | 750.5 |  |
| Total suspended solids | Pre | 34.69 | 0.00 |
|  | During | 45.71 |  |
|  | Pre | 34.69 | 0.00 |
|  | Post | 51.54 |  |
|  | During | 45.71 | 0.03 |
|  | Post | 51.54 |  |
| Total alkalinity | Pre | 343.8 | .39 |
|  | During | 340.9 |  |
|  | Pre | 343.8 | .48 |
|  | Post | 347.4 |  |
|  | During | 340.9 | .05 |
|  | Post | 347.4 |  |
| Chloride | Pre | 174.3 | 0.05 |
|  | During | 181 |  |
|  | Pre | 174.3 | 0.03 |
|  | Post | 184 |  |
|  | During | 181 | 0.17 |
|  | Post | 184 |  |
| Biochemical oxygen demand | Pre | 22.92 | 0.00 |
|  | During | 36.44 |  |
|  | Pre | 22.92 | 0.00 |
|  | Post | 39.19 |  |
|  | During | 36.44 | 0.15 |
|  | Post | 39.19 |  |

(*Continued*)

**Table 2.** (Continued)

| | Event | Mean | p-value |
|---|---|---|---|
| Chemical oxygen demand | Pre | 65.42 | 0.00 |
| | During | 56.56 | |
| | Pre | 65.42 | 0.00 |
| | Post | 54.05 | |
| | During | 56.56 | 0.07 |
| | Post | 54.05 | |
| Nitrate nitrogen | Pre | 6.9 | 0.58 |
| | During | 7.1 | |
| | Pre | 6.9 | 0.07 |
| | Post | 7.7 | |
| | During | 7.1 | 0.12 |
| | Post | 7.7 | |
| Total phosphorus | Pre | 3.35 | 0.07 |
| | During | 3.60 | |
| | Pre | 3.35 | 0.00 |
| | Post | 4.15 | |
| | During | 3.60 | 0.00 |
| | Post | 4.15 | |
| Organic phosphorus | Pre | 1.15 | .57 |
| | During | 1.21 | |
| | Pre | 1.15 | .30 |
| | Post | 1.29 | |
| | During | 1.21 | .51 |
| | Post | 1.29 | |

*TDS-total dissolved solids, TSS-total suspended solids, Talk-total alkalinity, Cl-chloride, BOD-biological oxygen demand, COD- chemical oxygen demand, NO3-N-Nitrate nitrogen, TP-total phosphorus, OrgP-organic phosphorus. $petitions paired t-test between various time points.

$paired t-test between various time points.

mean total coliform CFU/ml for river-water and–sediment was $94\times10^3$ and $48.7\times10^5$ respectively. CFU during- and post-bathing time-point was higher as compared to pre-bathing in river-water and in river-sediment (Table 3). *E. coli* CFU/ml was higher in river-water during-bathing which took place in monsoon and summer, while CFU was extremely high (about 200 to 300 times) in river-sediment for events in summer season.

## Antibiotic susceptibility to various antibiotic groups

The antibiotic resistance in *E. coli* from river-water and river-sediment for different antibiotics is shown in Fig 2.

The resistance pattern to various antibiotics was similar in *E.coli* sampled from river-water and river-sediment though the percentage of resistant bacteria was significantly higher in river-water ($p<.05$) than in river-sediment. However, considering the total number of coliform in CFU/ml in river-sediment, the total amount of resistance was significantly higher ($p = .04$) in river-sediment than in river-water. Resistance to polymyxin (colistin) group was not detected in any of the samples. Resistance to penicillin (27–40%) and cephalosporin (14–31%) was highest while it was lowest for tigecycline (0.3–4%) and gentamycin (1–3%). The MDR combinations having cephalosporin+quinolones+ penicllin and cephalosporin+quinolones

**Table 3. Mean number of *E. coli* collected during mass-bathing at different time-points from river-water and -sediment of river Kshipra, Central India (Friedman test).**

| Season | *Total *E. coli* in River-water | | | | **Total *E. coli* in River-sediment | | | |
|---|---|---|---|---|---|---|---|---|
| | Pre-bathing | During bathing | Post-bathing | *p*-value | Pre-bathing | During -bathing | Post-bathing | *p*-value |
| Monsoon 2014 | 9.4 | 7.3 | 24 | 0.00 | 1.7 | 3.8 | 2.2 | 0.36 |
| Winter 2014 | 0.86 | 0.60 | 0.91 | 0.00 | 2.7 | 2.6 | 4.4 | 0.26 |
| Summer 2015 | 1.0 | 3.6 | 12.0 | 0.31 | 93 | 330 | 7.4 | 0.26 |
| Summer 2015 | 3.7 | 51 | 2.3 | 0.84 | 1003 | 9590 | 116.6 | 0.71 |
| Winter 2015 | 0.83 | 0.49 | 5.2 | 0.84 | 7.8 | 18 | 25 | 0.71 |
| Winter 2015 | 0.19 | 0.95 | 0.43 | 0.03 | 18 | 17 | 32 | 0.26 |
| Winter 2016 | 0.15 | 0.25 | 0.09 | 0.56 | 17 | 32 | 9.8 | 0.71 |
| Winter 2016 | 0.15 | 0.44 | 0.77 | 0.03 | 26 | 29 | 9.0 | 0.09 |

*CFU- colony forming units,

**Values in x10$^3$

+ sulfonamides groups of drugs were more common than the cephalosporin+quinolone+-aminoglycosides or carbapenem combinations. Most of the isolates from all the sources showed resistance simultaneously to ceftazidime, cefotaxime, cefapime, ampicillin, tetracycline, and co-trimoxazole.

## Resistance of *E. coli* isolates between time-points of an event

The mean percentage of resistant *E.coli* was high in river-water for most commonly used antibiotics such as ampicillin ($p$ = .04), ceftazidime ($p$ = 0.00), cefotaxime ($p$ = 0.02), cefepime, nalidixic acid, ciprofloxacin, and nitrofurantoin during- and post-bathing while the percentage resistance to rarely used antibiotics in Indian clinical settings such as gentamicin, amikacin, tetracycline, sulfamethizole, imipenem ($p$ = 0.00) and meropenem ($p$ = 0.03)was more in pre-event period in both river-water and river-sediment (Table 4).

E. coli isolates from both river-water and -sediment belonged to phylogenetic groups A (69%), B1 (14%), B2 (3%), and D (14%).

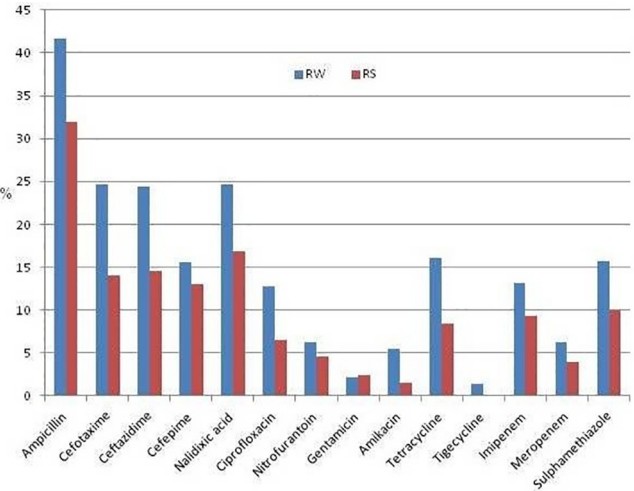

**Fig 2. Percentage of antibiotic resistance detected in *E.coli* isolated from river-water (RW) and river-sediment (RS) collected during eight mass bathing events during 2014–2016 from river-Kshipra, Central India.**

**Table 4. Antibiotic resistance detected in *E. coli* isolated at different time points of mass bathing events in river Kshipra, Central India.**

| Antibiotics Tested | River–water n (%) | | | River-sediment n (%) | | |
|---|---|---|---|---|---|---|
| | Pre-bathing n = 238 | During-bathing n = 269 | Post-bathing n = 253 | Pre-bathing n = 104 | During-bathing n = 101 | Post-bathing n = 116 |
| Ampicillin | 79 (33) | 94 (34) | 107(41) | 37(34) | 25(25) | 27(22)[g] |
| Ceftazidime | 53(22) | 30(11)[a] | 61(24) | 13(12) | 10(10) | 14(11) |
| Cefotaxime | 36(15) | 24(9)[b] | 36(14) | 14(13) | 08(8) | 11(9) |
| Cefapime | 17(7) | 16(6) | 21(8) | 10(9) | 06(6) | 09(7) |
| Nalidixic acid | 49(21) | 54(20) | 61(24) | 22(20) | 16(16) | 23(19) |
| Ciprofloxacin | 24(10) | 24(9) | 29(11) | 06(6) | 04(4) | 12(10) |
| Nitrofurantoin | 05(2) | 05(2) | 03(1) | 01(0.9) | 02(2) | 03(2) |
| Gentamicin | 03(1) | 03(1) | 00(0) | 01(0.9) | 01(1) | 02(2) |
| Amikacin | 02(1) | 03(1) | 00(0) | 0 | 0 | 01(0.8) |
| Tetracycline | 24(10) | 47(17) | 25(10) | 08(7) | 07(7) | 11(9) |
| Tigicycline | 01(0) | 00(0) | 00(0) | 0 | 0 | 0 |
| Imipenem | 26(11) | 09(3)[c] | 07(3)[e] | 01(0.9) | 0 | 04(3) |
| Meropenem | 11(5) | 04(1)[d] | 7(3) | 10(9)[f] | 0 | 05(4) |
| Sulphamethiazole | 38(16) | 36(13) | 27(11) | 10(9) | 12(12) | 09(7) |
| Colistin | 43(18) | 44 | 47(18) | 13(12) | 15(15) | 10(8) |

[a] *p* = 0.00,
[b] *p* = .02,
[c] *p* = 0.00,
[d] *p* = 0.03,
[e] *p* = 0.00,
[f] *p* = 0.00,
[g] *p* = 0.04

## Discussion

The physio-chemical and microbiological quality of river Kshipra deteriorated during- and after mass bathing events. Very high total coliform count/ml and *E. coli* CFU/ml was seen during-and post-bathing in river-water as compared to pre-bathing period. *E. coli* CFU/ml in river-sediment was much higher than in river-water. The antibiotic resistance was high with many MDR (9–44%) and ESBL producing (6–24%) *E. coli* in river-water as well as in river-sediment.

High total dissolved solid and free carbon dioxide were found during-bathing while high TSS, total and organic phosphorus, BOD and COD were detected in post-bathing period in river Kshipra. Thus, water quality parameters of river water deteriorate and do not remain fit to be used for drinking or household purposes [19–22]. The changes could be attributed to the use of soap, offering milk, flowers, oil etc. in river-water, dirt, and sweat of bathers', resuspension of sand and clay particles due to the discharge of wastes. Further, the highest worsening of water quality was also found to be correlated with the number of pilgrims that took bath during an event and most populated sites on the river bank [21].

Presence of indicator bacteria in water suggests poor water quality because of faecal contamination. Studies have reported that apart of release of domestic waste, leaching from septic tanks and farm wastes and mass bathing causes significant increase in faecal coliform load [19, 23, 24]. We found high number of *E.coli* during- and post-bathing in river-water and in river-sediment (Table 3). We interestingly, found that CFU/ml of total coliforms in river-sediment

is either same or more in pre-bathing period than during- and post-bathing which is in contrast to the finding for river-water. The increase in bacteria count in river-water during- and post-bathing thus could be attributed to mixing of river-sediment bacteria with the upper layer due to constant and heavy movement of water by movement of number of people during mass bathing. *E.coli* concentration of river-sediment was found to be re-suspended contributing to pollution of river-water [23]. Matson et al (1978) found mean faecal coliform counts in sediment to be 2,500 times greater than in overlying water [24]. It is therefore, suggested that microbial studies of bottom sediment be considered a part of water quality evaluations.

We also reported high number of *E.coli* in river-water and in river-sediment in events that took place in monsoon and summer. Bacterial count in river-water and -sediment varies with seasons being highest in summer [25, 26]. Gonzales et al also reported seasonal variation in coliform bacteria displaying an increased prevalence during the rainy season [27]. The levels of indicator bacteria is a sign of level of contamination and thus of unhealthy water with high potential of health risk [28].

Antibiotic resistant and MDR coliforms were between 20–38% in isolates from river- water and river-sediment which raise concern regarding risk of acquiring multiple antibiotic resistant genes from non-clinical settings via horizontal transfer to a wide range of bacterial species [3,4, 9,10,29,30]. Considering the number of *E.coli* (about 200 to 300 times higher) in river-sediment during summer season, the possibility of transfer of resistance genes may be alarming. Thus, river-sediment is appropriately considered as a natural reservoir of resistance genes [31]. Bacteria with highest level of resistance in river-sediment are the main reservoir of antibiotics and antibiotic resistance in environment [31, 29]. Our results, also showed a high percentage of commensal *E.coli*.

We noticed that commensal coliform are co-resistant to quinolones and cephalosporin which are mostly used in clinical settings, suggesting that environmental and clinical pathogens may have a common origin. The frequent use of quinolones group of antibiotics for treatment of enteric infections caused by gram-negative bacteria, could explain the high (25–40%), resistance of detected MDR *E.coli* against quinolones. We have previously shown that the frequency of resistance, co- resistance and resistance genes are high and similar in coliforms from humans and their environment [26,32]. Also, as previously seen the ESBL-producing *E. coli* isolates from river- water and -sediment belonged to phylogenetic groups A and B1 to which commensal *E. coli* are classified [26,32,33]. Thus, the emergence of MDR in *E. coli* and the other commensal coliform may also pose a risk for the emergence of new multi resistant pathogen.

The study has some limitations such as colistin resistance should have been performed and confirmed by MIC method and not by disk diffusion method. We, however, simultaneously performed mcr-1 gene detection by PCR. The disk diffusion method for AST is considered as a robust procedure though in some (5%) cases double and blurred zones were detected which were classified after repeat inoculation and confirmation by PCR. In addition, the zones of inhibition were checked by random blind assessment by another technologist. We, thus believe, that our findings of antibiotic resistance in river environment in India within a typical socio-behavioral context of religious mass bathing implies many important issues. Firstly, river acts as a medium for mixing pathogen and non-pathogenic bacteria from many sources like human, animal and environment; secondly, pathogenic bacteria may exchange antibiotic resistance genes with commensal; lastly, river-water acts as a natural reservoir and source of resistance genes for emerging pathogens. As during mass bathing in river Kshipra people from various parts of India, not only take a holy dip in the river but also use it for drinking, though high prevalence of coliform and the antibiotic resistance is an ideal setting for the acquisition and spread of antibiotic resistance genes and bacteria to distant places.

## Conclusions

Many physico-chemical properties of River Kshipra changed during mass bathing events. High levels of microbial contaminants that are resistant to multiple drugs were detected along the sampling sites of mass gathering in two years. These finding prompt for health risk of waterborne highly resistance enteric commensal transmission associated to the mass bathing in Ujjain city and in broader extent to Central India. The results emphasize the need for improved guidelines of surveillance of water quality and of antibiotic resistance during mass bathing festivals and should prompt policy makers to develop effective prevention and control strategies to avoid or minimize the risk of transmission of antibiotic resistance through water-borne coliform contamination.

## Supporting information

**S1 Data.**
(DOCX)

## Acknowledgments

The authors are thankful to management of R.D. Gardi Medical College, Ujjain, Salesh Chandran and H Shah, Professor of Microbiology, R.D. Gardi Medical College, Ujjain. We are also thankful to Girish Jain, Praveen Chouahn, Neha Sharma, Arjun Parihar, Vallabh Patidar, Richa Pandya and Mansingh Padiyar for sample collection and processing and Priyank Soni, Ankit Garg and Giriraj Singh for database management.

## Author Contributions

**Conceptualization:** Manju Purohit, Vishal Diwan, Ashok J. Tamhankar, Cecilia Stålsby Lundborg.

**Data curation:** Vishal Diwan, Vivek Parashar.

**Formal analysis:** Manju Purohit.

**Funding acquisition:** Cecilia Stålsby Lundborg.

**Investigation:** Manju Purohit.

**Methodology:** Manju Purohit, Vishal Diwan, Vivek Parashar, Cecilia Stålsby Lundborg.

**Project administration:** Cecilia Stålsby Lundborg.

**Supervision:** Cecilia Stålsby Lundborg.

**Validation:** Manju Purohit.

**Visualization:** Ashok J. Tamhankar.

**Writing – original draft:** Manju Purohit.

**Writing – review & editing:** Manju Purohit, Ashok J. Tamhankar.

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
