## [Decision Letter · Decision Letter 0]

25 Nov 2019

PONE-D-19-25307

Mass-bathing events in River Kshipra, Central India- influence on the water quality and the antibiotic susceptibility pattern of commensal E.coli

PLOS ONE

Dear Dr Purohit,

Thank you for submitting your manuscript to PLOS ONE. After careful consideration, we feel that it has merit but does not fully meet PLOS ONE’s publication criteria as it currently stands. Therefore, we invite you to submit a revised version of the manuscript that addresses the points raised during the review process.

Data on colistin susceptibility using MIC in agar or broth dilution is required as per updated CLSI guidelines

We would appreciate receiving your revised manuscript by Jan 09 2020 11:59PM. To enhance the reproducibility of your results, we recommend that if applicable you deposit your laboratory protocols in protocols.io, where a protocol can be assigned its own identifier (DOI) such that it can be cited independently in the future. For instructions see: http://journals.plos.org/plosone/s/submission-guidelines#loc-laboratory-protocols

We look forward to receiving your revised manuscript.

Kind regards,

Iddya Karunasagar

Academic Editor

PLOS ONE

Journal Requirements:

The project was funded by Swedish Research Council (grant no 521-2012-2889) and 2017-01237.

Include this sentence at the end of your statement: The funders had no role in study design, data collection and analysis, decision to publish, or preparation of the manuscript.

Additional Editor Comments:

Expert reviewers have pointed out the need to revise the manuscript updating information on antimicrobial susceptibility testing methodology and controls. Do you have data on colistin susceptibility using MIC method in agar or broth dilution? Please respond to reviewer comments point by point.

Reviewers' comments:

Reviewer's Responses to Questions

**Comments to the Author**

1. Is the manuscript technically sound, and do the data support the conclusions?

Reviewer #1: Yes

2. Has the statistical analysis been performed appropriately and rigorously? 

Reviewer #1: Yes

3. Have the authors made all data underlying the findings in their manuscript fully available?

Reviewer #1: Yes

4. Is the manuscript presented in an intelligible fashion and written in standard English?

Reviewer #1: Yes

5. Review Comments to the Author

Reviewer #1: 1. CLSI Performance Standards for Antimicrobial Susceptibility Testing are updated every year. Consequently, testing the antimicrobial susceptibility for Colistin requires MIC with agar or broth microdilution, and can no longer be reported using disk diffusion.

2. Kindly justify why this study carried out in 2014-16 is submitted for publication in 2019.

6. PLOS authors have the option to publish the peer review history of their article (what does this mean?). If published, this will include your full peer review and any attached files.

Reviewer #1: No

---

## [Author Response · Author response to Decision Letter 0]

23 Jan 2020

PONE-D-19-25307

Mass-bathing events in River Kshipra, Central India- influence on the water quality and the antibiotic susceptibility pattern of commensal E.coli

We are thankful to the Editor and reviewer for the comments and suggestions required for the improvement of the manuscript. Following are the reply to the various queries and concerns.

Editors Comments

Author's Reply: Formatting done as suggested.

2. Funding Statement: The funders had no role in study design, data collection and analysis, decision to publish, or preparation of the manuscript.

Author's Reply: Added the statement.

Author's Reply: Supporting files are added.

4. We note that Figure 1 in your submission contain [map/satellite] images which may be copyrighted. All PLOS content is published under the Creative Commons Attribution License (CC BY 4.0), which means that the manuscript, images, and Supporting Information files will be freely available online, and any third party is permitted to access, download, copy, distribute, and use these materials in any way, even commercially, with proper attribution. For these reasons, we cannot publish previously copyrighted maps or satellite images created using proprietary data, such as Google software (Google Maps, Street View, and Earth). 

Author's Reply: We have now replaced the previous figures and upload the map that is freely available in GADM database ( https://gadm.org/) and display only three sites of sampling. The figure 1 legend now shows all details. 

5. Expert reviewers have pointed out the need to revise the manuscript updating information on antimicrobial susceptibility testing methodology and controls. Do you have data on colistin susceptibility using MIC method in agar or broth dilution? 

Author's Reply: As mentioned in reply to reviewer in subsequent paragraph

Reviewer's Comments

1. Testing the antimicrobial susceptibility for Colistin requires MIC with agar or broth microdilution, and can no longer be reported using disk diffusion.

Author's Reply: Yes, we know that antimicrobial susceptibility for Colistin requires MIC with agar or broth micro-dilution. As pointed out in the next comment, we however did the analyses some time back and therefore we have included the explanation for the same in limitation as "The study has some limitations such as colistin resistance should have been performed and confirmed by MIC method and not by disk diffusion method. We, however, simultaneously performed mcr-1 gene detection by PCR", which to extent detected in colistin resistance. 

We missed to include the mcr-1gene detection in methodology in molecular method section, which we have added now.

2. Kindly justify why this study carried out in 2014-16 is submitted for publication in 2019.

Author's Reply: As the study is a part of bigger study conducted from 2014-17, which ran over to year 2018. Thus, other things came in between. All the laboratory work, however, was done simultaneously as data were collected and thus long ago before the new requirements for colistin susceptibility testing.

---

## [Editor Report · Decision Letter 1]

12 Feb 2020

Mass-bathing events in River Kshipra, Central India- influence on the water quality and the antibiotic susceptibility pattern of commensal E.coli

PONE-D-19-25307R1

Dear Dr. Purohit,

We are pleased to inform you that your manuscript has been judged scientifically suitable for publication and will be formally accepted for publication once it complies with all outstanding technical requirements.

With kind regards,

Iddya Karunasagar

Academic Editor

PLOS ONE

Additional Editor Comments (optional):

The reviewers comments have been addressed satisfactorily.
---

## [Editor Report · Acceptance letter]

20 Feb 2020

PONE-D-19-25307R1 

 Mass bathing events in River Kshipra, Central India- influence on the water quality and the antibiotic susceptibility pattern of commensal *E.coli*

Dear Dr. Purohit:

I am pleased to inform you that your manuscript has been deemed suitable for publication in PLOS ONE. Congratulations! Your manuscript is now with our production department. 

With kind regards,

on behalf of

Dr. Iddya Karunasagar 

Academic Editor

PLOS ONE